# North to South through a Post-Feminist Prism: Israeli Society as Reflected in Ora Shem-Ur's Fictional Detective Novels

Anat Koplowitz-Breier

Comparative Literature Department, Bar Ilan University, Ramat Gan 5290002, Israel;
anat.koplowitz-breier@biu.ac.il

**Abstract:** Ora Shem-Ur's detective series starring Ali Honigsberg established her as one of the early female pioneers in the new wave of Israeli detective fiction writers. In line with the current trend in post-feminist criticism towards analyzing the place of women within popular culture by looking at fiction as an agent of social change, this article suggests that the series not only addresses gendered topics but also other tensions and social exploitations of power within Israeli society. Shem-Ur thus provides a fascinating portrait of Israeli society in the 1990s, reflecting the way in which female detective fiction developed from light reading material into a social mirror presenting and addressing social changes and shifts in gender conception. Reading the series through a post-feminist lens, the article seeks to demonstrate how its themes of the relations between men, women, and power, and of economic corruption and politics, shed light on contemporaneous Israeli social issues.

**Keywords:** Ora Shem-Ur; detective fiction; Israeli literature; post-feminism; women authors; Israeli society

## 1. Introduction

The 1980s constituted a turning point in Israeli society and culture, with socialism and big government giving way to a neo-liberal belief in capitalism, the free market, and small government (Mautner 2011, p. 114). Moving away from a Eurocentric outlook, Israeli culture became increasingly Americanized and multicultural (Mautner 2001). This trend was reflected in Israeli literature (Hertsig 1998, p. 14), with the emergence of a new generation of "disillusioned" writers who had gone through the Yom Kippur War and ensuing political unrest (Oren 2009, p. 26). It also found expression in the development of non-canonical genres—including a new wave of detective fiction (Hever 2007, p. 241). Moving away from the "classic mysteries" and tight geometric patterns of whodunit fiction, the two pioneers of the genre in Israeli literature—Batya Gur and Shulamit Lapid, whose first detective novels were published in 1988 and 1989, respectively—employed the traditional crime form to probe the divisions within Israeli society (Abramovich 2010, p. 204). Despite being excluded from the Israeli literary canon until the 1980s (Fuchs 1987, pp. 6–7), women authors formed the vanguard of the women's prose fiction that emerged in Israel in the early 1980s (Feldman 1999, p. 10). As Gurevitz observes, this

> adopted another critical territory—one that touches on the gender discourse in general and on Israeli culture in particular. The very fact that women writers (Shulamit Lapid, Batya Gur and others) chose of all genres one considered peripheral was a show of defiance protesting the marginal role given to the female voice in the canonical stream of Israeli literature. (Gurevitz 2013, p. 295)

Although initially a peripheral genre, detective fiction has not only flourished but also gained a place of honor on the Israeli bookshelf. This is due to several factors: (a) the "gender discourse" to which it relates and the fact that from its very inception, it has featured female protagonists; (b) its deviation from the formulaic, puzzle-oriented mystery

type; (c) its expansion of focus to include characters and themes; (d) its use for reflection on/of social, political, and demographic rifts (Demko n.d.).

However, no other detective series—in particular with a female protagonist—have appeared for many years after those of Gur, Lapid, and Shem-Ur's. The latter's novels also gained much less scholarly attention than Gur's and Lapid's, with only one academic article addressing them (Berg 1994). Various reasons may account for this circumstance:

(1) Gur and Lapid were considered politically left or center, Shem-Ur liberal right;

(2) Lapid and Gur published their series with a prominent publishing house (Keter), while Shem-Ur with the very small feminist Noga;

(3) Gur's Oḥaion and Lapid's Badiḥi are Mizraḥim[1]; Shem-Ur's Ali Honigsberg is an Ashkenazi woman from a dysfunctional family.

In light of the fact that Israeli women detectives appearing on the (forensic) scene in the late 1980s and Israeli feminism was influenced primarily by the American movement, Israeli crime fiction must be understood in the context of worldwide trends in general and Anglo-American ones in particular (Feldman 2000, p. 269). Over the past 40 years, the genre of detective fiction, and especially women's detective fiction, has indeed shifted from mere light reading material to serving as a mirror of society (Irons 1995, p. x).

The early detective genre—in particular the American hardboiled detective —depicted a male detective with typical masculine traits. When women detectives figured in the stories, they were seldom perceived as having an individuality on a par with their male counterparts, either being old spinsters or depicted as using their feminine charms to compensate for their male-dominated profession (Slung 1975, p. xix). As literary critics note, most post-70s novels thus tend to frame women detectives in the context of gender power relations (Klein 1994). They thus "inherit the detective novel's traditions, but combine their elements into a new form" (Heilbrun 2002, p. 420). The fact that feminists spent more time reading than marching in protests in the 1980s (Munt [1994] 2005, p. 26) meant that feminist fiction in general and feminist detective fiction in particular became very popular. The majority of researchers thus tend to identify the 1980s as constituting a watershed in the field of female detective fiction.

Since then, Israeli feminism has begun exhibiting many of the features of Western feminism, most prominently in the move from "gender" to "genders" (Fuchs 2014, pp. 1–25), female detective fiction authors broadening the latter from a static concept pertaining to sexual difference into a "process of 'engendering'" (Fuchs 2009, p. 209). The acknowledgment of numerous feminist approaches opens up women's detective fiction to discussion from diverse perspectives—gender relations, violence and abuse, inter-female relationships, lesbian detective fiction, minorities and ethnic detective fiction, etc. In investigating crimes to uncover their perpetrators, detective fiction provides a fertile space for addressing such issues.

Ora Shem-Ur became part of this new wave with the publication of her series featuring Ali Honigsberg—*Murder in the Singles' Club* (1991), *Murder on Shenkin Street* (1992), *Murder in the Knesset* (1993), and *Murder at the Dead Sea* (1993). Deliberately portraying her protagonist as a thirty-plus unmarried woman as representative of all such Israeli women (Na'aman 1991, p. 73), she employs the feminist strategy of making invisible women visible—in conjunction with "post-feminist" principles, the latter variously (and contradictorily) signifying position, a type of feminism after the second wave, or a regressive political stance (Gill 2007, p. 147). I understand it herein to denote a "third space" that, rather than continuing or breaking from second-wave feminism, serves as a "middle ground" that allows women to be both feminine and feminists.

Examining the ideological and conceptual shifts Anglophone female crime fiction novels exhibit between the 1980s and 1990s, Canadian critic Sandra Tomc contends that many (Anglo-American) authors abandoned feminism in favor of articulating female experience in a post-feminist economy, finding their political rationale in contradiction (Tomc 1995, p. 47). As a new form of empowerment and independence, post-feminism is marked by individual choice, (sexual) pleasure, consumer culture, fashion, hybridism,

humor, and a renewed focus on the female body. It constitutes a new, critical way of understanding the changed relations between feminism, popular culture, and femininity (Adriaens 2009, n.p.).

## 2. Introducing Ali

Ali Honigsberg meets all the criteria Heilbrun sets out for female sleuths:

> Above all, these new women detectives prize their independence, offering no hostages to romance. Their living space remains their own, suited only to their needs. . . . They do not put men first in their lives, but they are rich in friends, sometimes men, and in lovers, sometimes women. (Heilbrun 2002, p. 421)

A "free spirit," Ali is single and freelance. Her work as a crime journalist for one of the larger papers and a translator of romantic novels affords her a lifestyle that enables her to move around at will, solving crimes across the country—from north to south and periphery to center. Although born in Kiryat Motzkin on the northern edge of Haifa, she lives in the heart of Tel Aviv, a city she adores: "Ali is in love with Tel Aviv. She has often thought that although she did not have the love of a man or a child—she at least loved the city' (*MSC* 81). She nonetheless retains some peripheral traits: "I have a wonderful jam, homemade. In Kiryat Motzkin where I grew up, everyone made jam" (*MS* 90).

Her attitude toward sexuality underscores her hybrid and post-feminism features. Despite considering herself a sexually liberated woman, free to have as many lovers as she likes—including married men—she has certain red lines she will not cross:

> I have no qualms about having an affair with a married man—I assume that when a married man cheats on his wife, he probably has reasons for it, and if not me, he'll find another object—but that's as far as I'll go. I would never agree to be the cause of a separation between a married couple. Never. That I refuse to have on my conscience, to be the cause of children growing up, because of me, in a home where Mom and Dad don't live under the same roof. (*MDS* 105)

She also condemns single parenthood—a rare phenomenon in the late 1980s and early 1990s in Israel—as denying children a fully functioning family (Roth 2016, n.p.). This philosophy derives from her biographical history—her mother abandoning her and her younger brother to an indifferent, rigid, and mentally abusive father. While her conservative approach towards family life appears to be at odds with her lifestyle, the post-feminism that enables women to be both feminine and feminist allows her to embrace the "importance of individual sexual pleasure and a choice of identity with both brain and body" (Schmidt 2015, p. 425). Subjects in their own individual right, post-feminist women can thus combine marriage, motherhood, and a career.

This range of options imparts a hybrid character to post-feminism. Ali exhibits her "conservative liberalism" again when pondering her trouble sleeping: "These are years of restlessness, anxiety attacks, of swings from the decision to find a spouse—no matter who—to deciding that it's better to live alone; of good days and bad days" (*MSC* 22). Combining a progressive, free lifestyle with conservative views, she is both a post-feminist and an integral part of the Israeli society of her time. While social changes began occurring in Israeli society during the late 1980s and early 1990s, in the Tel Aviv area in particular, the country remained largely conservative. Although "made-in-Israel," and, thus, someone with whom any Israeli reader—male or female—could identify (Na'aman 1991, p. 73), her liberalism thus makes her an outlier, Israeli culture lauding a steady job with a good salary (Sharabi and Harpaz 2007, p. 103) and holding a more traditional attitude towards marriage and family than most Western countries (Lavee and Katz 2003, p. 194). This hybridity may also represent the Israeli fusion of East and West (Smooha 2005, p. 441).

While claiming to be single "mostly by free choice" (*MDS* 107), Ali occasionally regrets her spinsterhood. Despite normally enjoying being "free as a bird" (*MSC* 60), she sometimes also entertains misgivings about her choice. Recognizing that the "sporadic thoughts" she has about Koby eventually marrying her—"and how can such thoughts be prevented?"—

are unrealistic (*MS* 94), she later observes: " . . . perhaps she should give up all hopes of getting married and enjoy what she has" (*MS* 95). This self-debate constitutes a further post-feminist trait, every woman needing to recognize her "own personal mix of identities" (Adriaens 2009, n.p.).

In another sense, Ali is an insider-outsider in Israeli society—an "outsider within" who inhabits a position populated by groups that, while having access to dominant cultural practices, are, for various reasons, unable to fully participate in them (Lenz 2004, p. 99). She thus belongs to a marginalized group whose position bestows them a unique standpoint from which to view mainstream culture (98).

At first glance, the murders in the series appear to reflect James' (2004, p. 12) four Ls—love, lust, lucre, and loathing—rather than constituting social criticism. The direct motives are relatively simple. Ali's friend Rivkah kills Giora in revenge for disappearing and leaving her an *aguna* (a woman whose husband refuses to divorce her or disappears but must remain married because not proven dead) and, thus, unable to remarry (*MSC*); Yigal kills Ronit in order to prevent her from using his parents' cellar because he needs it to cover his debts (*MSS*); Gali kills Ze'ev Ḥakim because he got her pregnant and then abused his niece (*MK*); and Dr. Giora Landau kills Mauricio in order to stop the latter from blackmailing him and destroying his marriage (*MDS*). While these reasons are personal and private, the detective-novel format—as opposed to literary or "mainstream" literature—enables Shem-Ur to shed light on strains and cracks within society. Like the popular feminist fiction of the 1980s, this genre addresses conventional viewpoints in order to impact cultural opinion, appropriating popular culture and genres in the process (Makinen 2001, p. 9). Participating in solving the mystery, the reader becomes aware of the evils of society (Scaggs 2005, pp. 74–75).

As a detective engaged in solving different types of mystery, Ali has access to diverse sectors of Israeli society. Detective work not only frequently involves reconstructing the complex movements of individuals through space and time (Kadonaga 1998, p. 414)—but the power to create a sense of place also serves as one of the criteria for regarding mystery writers as serious novelists (James 1983). Ali thus acts as a *flâneur* throughout the series (McDonough 2002; D'Souza and McDonough 2006; Koplowitz-Breier 2016, p. 96), the first volume being set primarily in Haifa and its environs (northern Israel), the second in Tel Aviv; the third in Tel Aviv and Jerusalem; and the last around the Dead Sea (southern Israel). Framing the books across both center and periphery enables Shem-Ur to address various social issues. Ali's hybrid nature also helps her fit in, thereby making it easier to identify the perpetrator. At the same time, she is an "outsider within", this unique perspective enabling her to reflect social disparities, her keen mind for solving mysteries also observing gaps and rifts in the social order. Herein, I shall address two subjects in particular—post-feminism and crony capitalism and corruption.

## 3. Gender and Power

While the crime that dominates the series is murder, sexual issues—prostitution (*MSC*), rape (*MS*), and the exploitation of young women by older men (*MK*)—also feature prominently. Like other women writers, Shem-Ur appears to espouse the tradition of using violence to make a "gendered protest" (Gavin 2010, p. 268). Associated with the feminine point of view, this approach is intensified by the fact that both author and protagonist are women. Female authors who choose (feminist) women protagonists often alter the male prototype by having their detectives speak from a woman's perspective, addressing the problems women face in modern society (Irons 1995, p. xii).

Ali explicitly claims to be a feminist—"'Me? I'm a feminist; I am a member of the association, I fight for equal rights for women and all that'" (*MSC* 55). In response to her question why she has never been appointed director, Yami, deputy director of the Knesset library, retorts:

> "There's always a man at the top of the ladder. I can point out ten secretaries who
> do the job of the ministers. So what? Does that make them ministers? To reach

the rank of deputy director general is an extraordinary achievement in our reality, and not just ours, all over the world."

"But you could at least fight!" protested Ali. (*MK* 41)

While Ali rails against gender discrimination, Yami is resigned to the "rules [of the game]," understanding that "doing gender" undermines the goal of dismantling gender inequity by (albeit inadvertently) perpetuating the belief that the gender system of oppression is impervious to change (Deutsch 2007, p. 107). The fact that this is the sole occasion in the entire series on which Ali supports a feminist act indicates her being a post-feminist which instead of asking feminist questions as a matter of course, "incorporates some of the insights about social life and power arrangements of feminist discourse without making them an explicit focus of analysis and debate" (Becker 2000, p. 399). In line with the trend prevalent in the 1980s, in which women contended with themselves rather than society and culture (Faludi 1991, p. 95), she thus deals with feminist issues privately rather than treating them as open, "political", or collective social problem.

Ali also exhibits an ambivalent attitude to gender relationships. On the one hand, she bemoans the way in which society makes different demands of men and women—"A man can seduce a woman, then another and another—and his success makes him macho. A woman who sleeps with men she likes, who easily succumbs to their overtures—is immediately considered a nymphomaniac" (*MSC* 166). On the other, she recognizes that "The days are gone when women were the ones who 'exploited' the men. There's no longer any discrimination between the sexes in this area" (*MSC* 196). In practical terms, she has no qualms about using her female/sexual wiles, especially when interviewing male suspects and informants, and/or sexual connections (maintaining an ongoing relationship with Eli Medan, a married police officer, throughout the series, for example).

This dual attitude is further reflected via the issue of prostitution—a thorny feminist question (Kesler 2002). A very thin line exists between the Tsimer sisters as professional streetwalkers and women such as Rivkah who use men for financial benefit in *Murder in the Singles Club*. Gavriela Tsimer asserts that both she and Rivkah are pursuing the same dream—i.e., finding a spouse for security. Thus, while many feminists denounce prostitution as part of the patriarchal system that exploits women's bodies, Shem-Ur recognizes that women also use men. However, the sisters specifically being adopted in order to make them prostitutes form an unusual case. It may thus be argued that rather than engaging in sweeping social criticism of prostitution—in Israel or abroad—Shem-Ur addresses the abuse of sexual power by men and women alike.

While *Murder in the Singles Club* deals with the subject of prostitution, *Murder on Shenkin Street* engages with the issue of rape. This topic has been extensively discussed by feminist scholars: liberal feminists considering it a "gender-neutral assault on individual autonomy" akin to other forms of assault and/or illegitimate appropriation while their radical counterparts identify it as originating in patriarchal constructions of gender and sexuality within the context of broader systems of male power (Whisnant 2017, n.p.).

The rapist in Shem-Ur's second volume targets Tel Aviv's Shenkin Street area. Although at the beginning of the book, the murder appears to be connected to this spree, the two eventually prove to be separate crimes. While portrayed as a very serious offence, not all the victims appear to perceive it as such. Olga, a seventeen-year-old Russian immigrant, for example, tells reporters that

she would never have made an issue of it, what's done is done, and why would she want to have anything to do with the police . . . "Maybe I'd have been more frightened if I'd been a virgin," Olga tried to explain why she was so unperturbed, "but he didn't do anything different from what my boyfriend does—or what the boyfriend I had in Russia did." (*MS* 105–6)

Olga treats being raped like any other form of sexual intercourse—i.e., as a patriarchal norm. This view accords with a documented trend: "There is a rise in the 'de-crime-ing' of

rape and sexual violence; there are new permutations of domestic violence, such as the rise of so-called 'boyfriend violence'" (McRobbie 2011, p. 184).

Ali takes a similar position:

> "What's done is done, it could happen to anyone. And you know what? It does happen to everyone . . . and if not actually, then just about. . . . You know how it is: a nice uncle. A neighbour. A friend. A boyfriend. Your boss at work. Believe me, I don't know a single woman who can't talk about rape or attempted rape. And not just one. There's nothing new under the sun, Ita, that's how men are. . . . That's the norm." (*MS* 142)

She nevertheless objects to Inspector Eliyahu Gur's distinction between rape and murder (*MS* 117), creating the impression that she regards both crimes as carrying the same weight.

Here, too, Shem-Ur thus appears to exhibit an ambivalent attitude: on the one hand, regarding rape as a sexual norm, both genders using and exploiting the other, and on the other, regarding it as a crime as serious as murder, with taking a person's body being just as abhorrent as taking a life. This dualism reflects the versatile approaches post-feminism adopts (Projansky 2001, p. 20).

In *Murder in the Knesset*, Shem-Ur exhibits a similar ambivalent attitude towards intergenerational relations and the exploitation of young women. When Ali finds out that Ze'ev has raped the young Gali, she observes that she is "ready to deal with all sorts of injustices—except against children. Against girls" (*MK* 17). However, later on, she expresses a different opinion: "For one thing, she doesn't think Gali is the innocent lamb her mother thinks she is. . . . Ali has little doubt that Ze'ev Ḥakim did not rape Gali, and that everything was done consensually' (*MK* 25).

Gali confirms the latter view, her attitude toward sex in general and her relationship with Ze'ev in particular being very different to her mother's assumptions. Despite her young age (sixteen), she asserts:

> I'm not very selective about sex. What's sex? Everyone has sex, but not everyone becomes famous. I always fancy celebrities. . . . So listen to what I'm telling you: Zevik was not that guilty. I have a friend, Avigail . . . so I told her about Zevik, and she said she was dying to get to know him . . . She said let's bet which one of us he'll make a pass at, so we sort of bet, and I sort of won, but it wasn't really fair, because I'd slept with him before I told Avigail about him. (*MK* 27)

Sex being a game for Gali—a way of gaining power over men—she does not blame Ze'ev. Ali is not shocked by this stance, arguing that Ze'ev's behavior is characteristically male: "It's true he likes young women, and the younger the better, but he's not alone in this: Ali lives in a society of men who identify with Ze'ev Ḥakim, if not in deeds, then in words" (*MK* 28–29). Despite Ali's cynicism, she relates to a problematic phenomenon in Israeli society—namely, relationships with thirty-plus men looking to sexually conquer teenagers.

This discussion suggests that Shem-Ur's primary concern lies with sex as power. In the series, both genders employ sex in this regard: Ali takes married lovers, her married neighbor Bracha and Julie Ḥakim following suit (*MK* 54, 108). Men use young women and women have younger lovers (*MK* 14). Both genders also regard sex and money as intertwined: "A male lover supports his female lover just as a female lover generously supports her male lover" (*MK* 104 [Eli Medan]). Virtually no difference exists between the genders where sex is concerned. However, although enamored of her free way of life, Ali is aware that society treats her differently, unmarried women not being regarded as "completely normal in most people's eyes" (*MS* 117). Shem-Ur thus approaches the balance of power between men and women in Israeli society with a critical eye.

## 4. Economic Corruption and Politics

While issues relating to the balance of power between men and women feature prominently in women's detective fiction (Kim 2012, pp. 2–6), Shem-Ur also addresses other

problems within Israeli society—including the decentralization of power whereby "variegated patterns of the use of power and political activity are created" (Herzog 2005, p. 86). Since the late 1970s, Israeli society has been plagued in particular by crony capitalism or political corruption.

Political and social scientists frequently speak of three types of corruption—white, grey, and black (Werner 2002, p. 211). The level of each of these depends upon the perception of the activity by public and public officials. When both sides agree that a deed should be condemned and punished or left unsanctioned, it is classified as black or white, respectively. Where disagreement exists, it falls into the grey category. In post-1980s Israel, the boundaries between these three forms of corruption became blurred, allowing corruption to "mature along with the maturation of the political system" (Werner 2002, pp. 216–17).

Written at the beginning of the 1990s, Shem-Ur's series was published when political corruption had become well established within Israeli politics, being conspicuous for its ubiquity. Matskin divides Israeli political corruption from the pre-state period up to the present into four periods, the fourth (1977 onwards) being the "institutionalization of the phenomenon of political corruption at the same time as its recognition as an evil that crosses political boundaries" (Matskin 2012, p. 30). According to Navot, "Corruption—and the talk about it—were among the most important factors in the defeat of the Labour Party in the elections to the Ninth Knesset in May 1977" (Navot 2012, p. 95).

The economic changes that ensued in the wake of the latter not only failed to counter the phenomenon but entrenched it even further: "If there was hope in the public that the change of government would lead to a change in moral norms, proper administration, and integrity in the political system and in the public sector in general, these hopes were proven false" (Navot 2012, p. 92). The shift from socialism and big government to a neo-liberal capitalism, free market, and "small" government transformed Israel's economy (Mautner 2011). This led to even greater political corruption, the power–money nexus representing a major factor in government corruption (Lev-Ari 2011, p. 32).

Shem-Ur's novels portray several aspects of the corruption and power–money relationships rife within the Israeli economy of the day. In the second book, *Murder on Shenkin Street*, the murder victim—Ronit Rav'on—is the daughter of Ali's lover Koby, a wealthy businessman who, while generous, kind, relaxed, charming, considerate, and understanding in her eyes is also

> willing to trample, kick, mislead, lie, cheat—otherwise he wouldn't have become one of the most influential people in the country, both in the economic, public—and political—sectors. He's the type that tries to keep as far away as possible from the limelight and public positions. The type that prefers to pull the strings. To run the politicians. To get economic laws passed in the Knesset—laws that rather than benefit the state benefit him. (*MS* 11–12)

Using money to influence both politicians and the private sector, Koby's modus operandi is characteristic of corruption (Amundsen 1999, p. 2). The only people Koby cannot manage are in fact his older children—Ronit, who takes the radical step of changing her family name to X to disassociate herself from her father, and Sami, who joins the IDF against his father's wishes. When Ali speaks to Koby about Sami, he claims: "He enlisted. Without my knowledge. My son doesn't have to serve in Gaza. . . . What I contribute to the state is enough to exempt my son from service" (*MS* 40). Koby considers himself superior to others because of the influence he wields in high places.

Yizhar's father Tzadok Nesher is Koby's mirror image. As a politician, his public life is "an open book. He didn't betray or embezzle but led an impeccable life. . . . Tzadok Nesher was known for his asceticism" (*MS* 196). Although keeping his personal life as private as possible, some unsavoury aspects were nevertheless common knowledge. These indicated that, where his family was concerned, he was not above using his power to his own advantage, extricating his eldest son from a police investigation into his younger son's death and smuggling him abroad under a false name, and calling in some favours to let

Yizhar sit his officers' training course exams again (*MS* 237). However, by not abusing his political power to bring the state down or attempting to change laws for his own benefit, he differs from Koby.

In the final book in the series, *Murder at the Dead Sea*, Shem-Ur portrays the wealthy as wielding influence over politicians. Explaining how he obtained planning approval to build a new hotel, Sigmund Brodetti observes:

> The permit to build the new hotel gets delayed, God knows when those clerks will put their signature on it, there's a different minister, so everything is stuck. Now I have to start courting the new minister. With the previous minister I got along like a house on fire. Everyone knew you had to take him on an interesting trip, put him up in a five-star hotel and place a limousine at his disposal—and the license was yours. (*MDS* 147)

Brodetti makes no effort to disguise his willingness to engage in bribery. When he claims to be a sensitive man, Ali represents him as corrupt:

> Sigmund Brodetti, the survivor who made a fortune from brothels in Frankfurt, who made his millions by exploiting and deceiving and abusing young girls who came from villages to the big city to earn something to eat, a man who doesn't hesitate to bribe any official, from the lowest to the highest. (*MDS* 151–52)

The ties between big money and power are perhaps most prominent in the third novel, in which the corruption has become political. Married to Julie Shoham, the daughter of a millionaire tycoon for whom he used to work, Ze'ev Ḥakim asserts that the marriage was a love alliance (*MK* 62). However, in reality, it served both partners' needs (*MK* 120–21). Ḥakim not only uses his father-in-law's money and influence to buy his way into the Knesset but also exploits his own political power to help his father-in-law in return:

> "And when you were a member of the Finance Committee, you succeeded— all in the name of ideology, of course—in rezoning these agricultural lands for construction—and it's pure chance that your father-in-law just happened to purchase those very rezoned citrus groves not long ago." (*MK* 63)

When Ze'ev argues that other such deals were made, the newspaper reporter retorts: "But for some reason, the State Comptroller found fault with this one" (*MK* 63). Ali thus concludes: "Apparently, a lot of bribe money exchanged hands at the time—although the ethics committee Ze'ev appeared before found no proof of corruption and ruled in his favour" (*MK* 63).

This activity meets Khan's definition of "political corruption" as "behaviour which deviates from the formal rules of conduct governing the actions of someone in a position of public authority because of private-regarding motives such as wealth, power or status" (Khan 1996, p. 12). During the investigation, Ali wonders whether Max Shirazi could have murdered Ḥakim after Shirazi's secretary reveals that her boss was working to change the law concerning the profits from the discovery of oil:

> Max worked on Ze'ev to get him into his network, so he wouldn't be able to say "no" when really serious matters arose. Max invested a lot of money in all those geologists who tested the land for him, and in the end, he bought the area where everyone said there must be a lot of oil. But Max Shirazi is not one to invest to make the government rich. So he worked on changing the law. . . . The entrepreneur invests—the entrepreneur expects to turn a profit. (*MK* 63)

Although Ze'ev is killed because of his sexual appetite rather than due to political corruption, the fact that Shirazi expects him to pressure the committee into giving him and others benefits in order to change the law and thereby earn him greater profit from state property heightens the issue of bribery and the unsavory dealings between the business and political sectors. His acts resemble Koby Rav'on's in *Murder on Shenkin Street* and to what Siegfried Brodetti confesses of doing in *Murder at the Dead Sea*. Ze'ev's deeds reflect the other side of the coin—political corruption. The novels thus clearly illustrate

the blurring of the three categories corruption, their depiction of Israeli society as tainted with corruption—particularly political—accurately reflects the rise of the phenomenon and public awareness of it. Ali's perception view of it indicates Shem-Ur's own view that it must be critically assessed.

## 5. Conclusions

Ali conforms to the paradigm of the female detective: few are married, 98 percent being single, divorced, or widowed, and a minority actively hunting husbands (Mizejewski 2004, p. 12). Despite being single and adopting a sexually liberated lifestyle, Ali is a mixture of liberalism and conservatism. Hereby, she represents the post-feminist hybridity that allows women to be both feminists and feminine. As a post-feminist, Ali reflects the treatment of relationships between men and women as part of social rather than feminist discourse, with both men and women engaging in the exploitation of power. Rather than distinguishing between rape, sexual harassment, and sex in an unequal relationship, or between prostitution and sexual exploitation of men by women for economic needs and vice versa, Shem-Ur regards all these phenomena as belonging to the same category.

Shem-Ur's use of the detective genre to portray the problematic aspects of Israeli society is in line with other Hebrew women detective writers—from pioneers such as Shulamit Lapid to contemporary novelists such as A. Gefen and Y. Shelach—whose protagonists are women. Ali's hybrid nature sharpens her detective skills. Knowing all the murderers personally, she must distance herself from them in order to solve the crime, exiting rather than entering other's lives (Berg 1994, p. 286). Her independence is an asset, enabling her to look at people and society from an outsider's perspective.

The detective genre also embodies the active-passive act of reading, mixing "writerly" and "readerly" texts (Barthes 1974, p. 4). In employing such a popular genre, Shem-Ur makes use of its capacity to encourage readers to solve the mysteries alongside the detective via the reading process, thereby making them aware of the evils of society presented in the text (Scaggs 2005, pp. 74–75). This dual process leads to the "apprehension" of both the criminal and the social injustices underlying the crime.

**Funding:** This research received no external funding.

**Institutional Review Board Statement:** Not applicable.

**Informed Consent Statement:** Not applicable.

**Data Availability Statement:** Not applicable.

**Conflicts of Interest:** The author declares no conflict of interest.

## Note

1      Mizraḥim: Jews who were born in Arab or Arabic-speaking countries.

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
