# Peer review of "North to South through a Post-Feminist Prism: Israeli Society as Reflected in Ora Shem-Ur’s Fictional Detective Novels"

_humanities, doi:10.3390/h11060133_

Round 1

Reviewer 1 Report

The article “North to South through a Post-Feminist Prism: Israeli Society as Reflected in Ora Shem-Ur’s Fictional Detective Novels” offers an insightful commentary on Shem-Ur’s detective novels from the 1990s as a post-feminist, social critique of Israeli society. Reading the novels through the lens of post-feminist theory, the author shows how Shem-Ur novel follows in the Anglo-American tradition of female detectives and that of contemporaneous Israeli female detective novels. In this paradigm, the post-feminist detective occupies a third space that embraces contradiction and allows her to be both feminine and a feminist. The article’s author supports this view with numerous examples of the lead detective’s ambivalent position in society, as a mix of conservative and liberal, insider and outsider, and with dual positions on issues of gender and power issues, including gender discrimination, prostitution and rape. The author further states that Shem-Ur’s novels use the detective genre to shine a light on widespread economic and political corruption in Israeli society. The article concludes by reiterating that Shem-Ur’s detective exemplifies post-feminist hybridity and that this unique position sharpens her detective’s gaze and thereby her ability to shed light on societal issues of gender, politics and power.

This article has numerous strengths, beginning with its topic. The author’s analysis of Israeli female detective novels sheds light on an overlooked area of literature and can contribute to a broader understanding of socially critical detective novels beyond the Anglo-American framework and those written by females more specifically. The post-feminist framework is particularly clear and is well-developed and illustrated by multiple concrete analyses from Shem-Ur’s novels (pp. 2-6). Also clear and convincing are the examples of Shem-Ur’s critique of economic and political corruption (pp. 6-8), although the connection between these, post-feminism and the detective genre are less obvious and only alluded to the conclusion.

The main area where the article could be improved is in the theoretical framework regarding the detective genre. The most consequential comments on this genre are in the introduction and conclusion, with the conclusion having the strongest and clearest statements on the importance of the genre for Shem-Ur’s aims. What is missing is a more consistent and developed discussion of the genre’s impact throughout the analysis. For example, the section that discusses the main character Ali (p. 2-3) focuses on her hybrid, post-feminist identity and then abruptly ends with an unconnected paragraph (lines 128-140) that summarizes the types of murders in the novels and claims that the detective genre format, because it is not a mainstream genre, allows Shem-Ur to “shed light on strains and cracks within society” (p. 3, line 137). This is followed by the statement that as readers participate in the mystery, they become aware of social ills. While these claims are supported by citations of secondary texts provided in parentheses, the author doesn’t develop these ideas and doesn’t link them clearly to either the post-feminist discourse or to the two subjects that follow (namely the type of critique). This would be a great place in the article to develop the connection between detective fiction and this article’s specific focus. This could be done with a stronger, more organic link to the preceding discussion of the detective’s character and to the following sections. Similarly the two sections that make up the bulk of the analysis of the novels (pp. 4-8) should also include more explicit ties between the novels’ structure (as detective novels) and the critique of gender and power and economic corruption and politics. Rather than just quoting or referring to secondary literature that proclaims the active role of the reader in detective fiction and the genre’s tendency to showcase social critique the author should show how Shem-Ur does so.

A second place where the article’s argument could be strengthened is in emphasizing its contribution to existing scholarship. While the abstract claims this series establishes Shem-Ur as “one of the early female pioneers in the new wave of Israeli detective fiction writers” the article itself does more to link her to contemporaneous trends, such as those in Anglo-American literature and that by other female Israeli crime authors such as Batya Gur and Shulamit Lapid, than to show her uniqueness. The author should establish at key points throughout the article (such as in each section) what makes Shem-Ur stand out as a pioneer rather than what makes her the same. Maybe a more explicit reference at the beginning to how rare she, Gur and Lapid were then as opposed to the (presumably) many who followed? This article does make a great contribution to an understanding of the global detective genre from a less emphasized country and this should be highlighted.

A third improvement would be a slight modification in the use of secondary texts. The author has many valuable sources, but seems to borrow a line or two from each rather than picking a few dominant experts and consistently using these. This was most obvious (and potentially problematic) in the section on economic corruption and politics (pp. 6-8) which contains definitions of power and political corruption from Herzog, Amundsen, Khan, and Werner. Each one emphasizes different nuances that shift the view of what the social ills are that Shem-Ur critiques. Particularly Werner’s ideas of white, grey, and black corruption is only introduced at the end of the section (p.8) and thus seems to be an “add-on” after the text analyses. I would recommend using less sources and definitions and focusing on one or two.

The article is clear, well-written and supported with many references to theoretical texts. It is also well proofread and is free of typographical errors. A few suggestions on style are, however, to limit the number of long, run-on sentences (ex. Lines 43-46, 109-113, 130-135). Also, varying the sentence structure more would make the ideas clearer and stronger. Most obvious is the preference for beginning sentences with gerunds and or subordinating conjunctions: i.e. “Israeli women detective appearing on the (forensic) scene ….and Israeli feminist being influenced primarily by the American movement, Israeli crime fiction…” (lines 43-45) followed by “Concretizing the feminist argument… (line 46); or “Combining a progressive, free lifestyle with conservative views, she is both…” (lines 105-106), “While social changes began occurring….” (line 107), “Although ‘made-in Israel’” (109), “While claiming to be single ‘mostly by free choice,’” (115), “Despite normally enjoying being ‘free as a bird,’” (116), “Recognizing that the ‘sporadic thoughts’ she has…” (117). Rephrasing some of these will improve the overall (good!) writing style. Another suggestion would be to reword the transition on p 3 (lines 141-42) to be consistent with the subject headings that follow.

Author Response

I read thoroughly the reviewer’s suggestions and I did the following revisions:

  1. I added material about the theoretical framework regarding the detective genre and its importance to Shem-Ur’s novels. I also developed the connection between detective fiction and this article’s specific focus (and with Ali, its detective, hybridity).
  2. I didn’t totally agree to the reviewer’s claiming that in the bulk of the analysis of the novels I was “just quoting or referring to secondary literature that proclaims the active role of the reader in detective fiction and the genre’s tendency to showcase social critique the author should show how Shem-Ur does so.” Nevertheless, I added some explanation regarding Shem-Ur’s way of using Ali’s critical point of view as a tool giving her own views.
  3. I added information about the development of the feminist detective fiction in the Anglo-American world and its way of effecting the Israeli pioneers. I also showed the uniqueness of Shem-Ur’s novels and tried to explain why she was not regarded as one of the pioneers, also I claimed she is.
  4. I don’t agree with the reviewer’s statement: “The author has many valuable sources, but seems to borrow a line or two from each rather than picking a few dominant experts and consistently using these.” I think the different aspects (especially concerning the corruption issue) are relevant to Shem-Ur’s writing. Nevertheless, I think his remark concerning Wilner typology is right, and I did some revisions regarding it.
  5. As for his English suggestions, I have addressed the comments regarding stylistic features and altered sentences where relevant.

Reviewer 2 Report

The writer presents sime interesting insight with regard to post-feminist treatment of 1980-1990s female detectives. She/He neatly connects between the hendered represntations of her protagonists and the socio-economic climate in Israel in the above-mentioned period.

The style, grammar, punctuation, sentence structure- need very serious improvement. Some sentences agr grammatically incorrect (see lines 22-23 OR lines 43-45 OR lines 167-169).

THe writer does not use punctuation correctly, either omitting commas, or adding them when unnecessary.

She/He uses the word "thus" 16 times in the text, mostly incorrectly, when there is no relationshiop of cause and effect.

Line 79 - "although" - redyndant - no contrast there

She/he often uses incorrect terms. Abstract - line 13 - she/he refers to "women" and "men" as THEMES. These are not themes.

Line 131 - she/he needs to explain the term "aguna". Many people, especially non-Jews, do not understnad this term.

To sum up, while the topic and some ideas are interesting and contribute new insights to the the literary feminist discourse, the article should be carefully edited in order to be published.

Author Response

Since all of the reviewer’s suggestions regarded the English style of my article, I’ve sent it to my professional language editor (since I am not a native English speaker, I always uses her services). Her reply was: “The comments of the second reviewer have been noted but appear unjustified. They have thus not been addressed specifically.”

Round 2

Reviewer 2 Report

The writer did not correct any of the mistakes (grammatical, stylistic and bibliographical ) that I suggested in the previous review, claiming that her editor said there were no problems withe the above mentioned problems.

It is a pity.

I am attaching more comments on the author's second version

I hope that this time he/she will relate to these comments

Author Response

  1. In the abstract: I revised the places the reviewer’s suggested.
  2. I added the page numbers in most of the places the reviewer asked. In the places I didn’t it is because the reference as a whole deal with the issue it refers to.
  3. I did most of the grammatical and punctuation changes the reviewer asked for.
  4. I added a footnote explaining what Mizraḥim are as the reviewer requested.
  5. I revised most of the sentences the reviewer claimed were unclear.